# The Role of Advance Care Planning in Cancer Patient and Caregiver Grief Resolution: Helpful or Harmful?

**DOI:** 10.3390/cancers13081977

**Published:** 2021-04-20

**Authors:** Francesca Falzarano, Holly G. Prigerson, Paul K. Maciejewski

**Affiliations:** 1Cornell Center for Research on End-of-Life Care, Weill Cornell Medicine, New York, NY 10021, USA; fbf4001@med.cornell.edu (F.F.); pam2056@med.cornell.edu (P.K.M.); 2Department of Medicine, Weill Cornell Medicine, New York, NY 10021, USA; 3Department of Radiology, Weill Cornell Medicine, New York, NY 10021, USA

**Keywords:** grief, advance care planning, advanced cancer, family caregivers

## Abstract

**Simple Summary:**

Grief is a common emotion felt by advanced cancer patients and their family caregivers, yet little is known of the relationship between grief in patients and caregivers, how grief in patients and caregivers changes as patients get closer to death, and the way advance care planning (ACP) relates to grief in both members of this “care pair.” In a sample of advanced cancer patients and caregivers, we found their grief tended to be synchronized and that, on average, patients’ grief remained stable whereas caregivers’ grief declined. Further, results revealed that completion of a living will (LW) for the patient increased levels of patient grief, while completion of a do-not-resuscitate (DNR) order decreased levels of caregiver grief. Results suggest that grieving may be synchronized between patients and caregivers and that while ACP may promote grief resolution for family caregivers, it is evocative of grief for patients.

**Abstract:**

Cancer patients and their family caregivers experience various losses when patients become terminally ill, yet little is known about the grief experienced by patients and caregivers and factors that influence grief as patients approach death. Additionally, few, if any, studies have explored associations between advance care planning (ACP) and grief resolution among cancer patients and caregivers. To fill this knowledge gap, the current study examined changes in grief over time in patients and their family caregivers and whether changes in patient grief are associated with changes in caregiver grief. We also sought to determine how grief changed following the completion of advance directives. The sample included advanced cancer patients and caregivers (*n* = 98 dyads) from Coping with Cancer III, a federally funded, multi-site prospective longitudinal study of end-stage cancer care. Participants were interviewed at baseline and at follow-up roughly 2 months later. Results suggest synchrony, whereby changes in patient grief were associated with changes in caregiver grief. We also found that patients who completed a living will (LW) experienced increases in grief, while caregivers of patients who completed a do-not-resuscitate (DNR) order experienced reductions in grief, suggesting that ACP may prompt “grief work” in patients while promoting grief resolution in caregivers.

## 1. Introduction

Although a common, nearly universal experience among advanced cancer patients and their family caregivers, grief prior to a loved one’s death remains an understudied topic. Grief in this context has been described as the anticipation of one’s own or a loved one’s future death, coupled with the navigation of prior and ongoing losses as a result of a terminal illness [1,2,3]. It is often unrecognized or mistaken for mental disorders such as anxiety, posttraumatic stress, or depression [1,2,4,5,6]. Evans [7] describes grief during caregiving as a “reaction to a multiple loss situation” rather than a reaction “to the loss incurred by the actual death” (p. 163), highlighting the multifaceted experience of grief that takes place in advance of a physical death. These losses include the loss of healthy functioning, the impending loss of life, the loss of a close, possibly self-defining relationship with the patient, diminishing social connections as the care role intensifies, and loss of control as family members confront an inability to prevent the patient’s impending demise [5,8,9,10,11].

In contrast to caregiver grief in bereavement, grief near the end of a patient’s life is experienced by both the patient and the caregiver, as both are forced to confront the patient’s loss of functioning, autonomy, future goals, and, ultimately, their life [12]. For both patients and caregivers, symptoms of grief include disbelief, anger, preoccupation with thoughts of the loss, and longing for the patient’s former, healthier self [13]. Research has also found grief in end-of-life contexts to be associated with depression, anxiety, poor coping and impaired quality of life, worse end-of-life decision-making, and lack of death preparedness [2,10,14,15]. Additionally, for caregivers, heightened grief near the end of patients’ lives can lead to poor bereavement outcomes–including Prolonged Grief Disorder, itself one of the strongest predictors of suicidal ideation in bereaved caregivers [16,17,18,19,20]. Despite a growing body of research examining psychosocial correlates of patient and caregiver grief near the end of patients’ lives, little is known of the course of grief for patients and family caregivers as patients approach death.

Bowlby and Parkes [21,22,23] were among the first to propose stages of grief resolution. Rooted in attachment theory, their stages of grief included an initial reaction of numbness and shock, followed by protest and yearning, despair and hopelessness, and ultimately acceptance or reorganization. Building on this theory, the widely popular but contentious stage model of grief [24] has received empirical support in research on bereaved subjects [25,26] but has not been corroborated among dying patients as Kubler-Ross [24] had proposed, nor among caregivers of patients prior to the patient’s death. This model suggests that over time, the dying patient’s grief diminishes as patients follow a sequence starting with disbelief and then followed by anger, bargaining, depression, and ultimate acceptance. Yet, it remains to be empirically demonstrated that such stages are consistent with the course of grief experienced by advanced cancer patients and their family caregivers as patients approach death. Further, few, if any, studies have investigated changes in patient and caregiver end-of-life grief over time and how end-of-life medical decision-making behaviors, such as advance care planning (ACP) (e.g., completion of living wills (LW) and do-not-resuscitate (DNR) orders) affect patient and caregiver grief trajectories.

While studies have demonstrated the interdependency of mental health between patients and caregivers [27,28], research has yet to examine how the experience of grief near the end of patients’ lives changes in patients and caregivers over time, as well as factors that influence changes in grief. Further, evidence suggests that advance directives, such as completion of a DNR and LW, are associated with less aggressive treatment and better quality of life in patients, as well as better death preparedness, reduced decisional burden, and lower distress in caregivers at the end-of-life [29,30,31,32,33,34,35,36]. Despite associations between advance care planning and better psychosocial outcomes near the end of patients’ lives, the influence of advance directive completion on grief in patients and caregivers and whether changes in grief run parallel to each other among patients and caregivers has not been empirically examined.

Understanding how the grief experience changes over time in patients and caregivers may inform the development of comprehensive interventions that address the needs of both patients and families confronting a terminal prognosis. Thus, the aims of the current study are three-fold: (1) to evaluate changes in grief in terminally ill patients and their family caregivers over time as the cancer patient approaches death, (2) to examine the synchronicity of grief changes in patients and caregivers over time, and (3) to investigate the role of ACP, specifically through the completion of advance care directives, such as LWs and DNR orders, on changes in grief for both patients and caregivers.

## 2. Materials and Methods

### 2.1. Study Sample

The sample for the present study (*n* = 98 patient-caregiver dyads) is a subsample of patient and caregiver participants in Coping with Cancer III (CwC III), a multi-site, prospective, longitudinal cohort study of advanced cancer patients and their caregivers funded by the National Institute of Minority Health and Health Disparities (MD007652; MPIs: Maciejewski/Prigerson) to examine Latino/non-Latino disparities in ACP and end-of-life care. Thus, this is a secondary analysis of these data for the purpose of examining how grief changes in patients and their family caregivers as the patient approaches death and the influence of advance care planning on grief resolution of both patients and their family caregivers. CwC III participants were recruited between November 2015 and May 2019 at seven institutions: Memorial Sloan Kettering Cancer Center (New York, NY, USA), Columbia University Medical Center (New York, NY, USA), Northwestern University Robert H. Lurie Comprehensive Cancer Center (Chicago, IL, USA), Rush University Medical Center (Chicago, IL, USA), University of Texas-Southwestern (Dallas, TX, USA), University of Texas at El Paso (El Paso, TX, USA), and University of Miami Health System (Miami, FL, USA). IRB approval was obtained from all participating sites. Informed consent was provided by all study participants.

Patients were eligible to participate in CwC III if they had been diagnosed with a locally advanced or metastatic gastrointestinal, lung, or gynecological cancer and had experienced disease progression on at least first-line chemotherapy or, for some specific (e.g., colorectal, ovarian) cancers, had experienced disease progression on at least two lines of chemotherapy. Thus, in general, eligible patients had incurable cancers and limited (i.e., months, not years) life expectancies. Caregivers were eligible to participate if participating patients identified them as those (e.g., family members or friends) who provided most of their informal care. Patients and caregivers were excluded from participation if they were under 21 years of age, not fluent in either English or Spanish, severely cognitively impaired (as indicated by a score of less than 6 on the Short Portable Mental Status Questionnaire [37]), or judged by research staff members to be too weak or ill to complete study interviews.

In CwC III, trained research staff interviewers gathered information directly from patient and caregiver participants longitudinally in up to three separate structured interview assessments per participant. Baseline assessments were conducted at study entry, and first follow-ups were conducted approximately 2-months on average post-baseline, and second follow-ups were conducted up to 1-year post-baseline (depending on patients’ statuses). Patient and caregiver participants were compensated with 25 USD per interview.

The sample for the present analysis (*n* = 98 patient-caregiver dyads) consists of CwC III patient and caregiver participants who completed their first follow-up interview assessments within 4 months of their baseline assessments and provided answers to questions evaluating their grief. Defining the study sample in this way allowed us to evaluate our two outcomes of interest, i.e., temporal changes in patient and caregiver grief, within a well-defined time frame.

### 2.2. Measures

#### 2.2.1. Sociodemographic Characteristics

During their baseline interviews, patients and caregivers provided information regarding age, sex, race/ethnicity, education, marital status, insurance status, and income. Caregivers were asked to report on their relationship to the care recipient, as well as whether they currently reside with the patient.

#### 2.2.2. Temporal Changes in Patient and in Caregiver Grief

We assessed grief in patients and caregivers during their baseline and first follow-up interviews using the following three items from the Prolonged Grief-12 (PG-12) Scale [2,38]: 1. “How often have you felt yourself longing or yearning (for you/the patient) to be healthy again?”; 2. “How often have you felt shocked, stunned, or dazed by (your/the patient’s) illness?”; 3.“Do you feel emotionally numb since (you/the patient) was diagnosed?”. Response options for each item ranged from 1 = not at all to 5 = several times a day. In the present study sample, this abbreviated, three-item grief scale had acceptable reliability among patients (Cronbach’s α = 0.60) and caregivers (Cronbach’s α = 0.65). For each participant, we calculated a change in grief score by subtracting the average score for these three items at baseline from the average score at follow-up.

#### 2.2.3. Advance Directives

At baseline, patients were asked if they had completed a DNR order, an LW, and/or designated a healthcare proxy. Responses for each item were coded as 1 = yes and 0 = no.

### 2.3. Statistical Analysis

Participant sociodemographic characteristics and patient use of advance directives are described using frequencies and percentages. Temporal changes in patient and caregiver grief are described using means and standard deviations. Bivariate associations between participant characteristics and changes in patient and caregiver grief were examined using Pearson’s correlations. Multiple linear regression models were used to estimate independent effects of participant characteristics, including patient use of advance directives, on two outcomes, i.e., changes in patient and in caregiver grief. In the regression model for each outcome, participant characteristics found to be either significantly (*p* < 0.05) or slightly non-significantly (*p* < 0.10) associated with the outcome bivariately were included as independent variables. Changes in caregiver grief were considered a potential predictor of changes in patient grief, and vice versa.

Statistical analyses were conducted using SAS statistical software, Version 9.4 (SAS Institute Inc., Cary, NC, USA). Inferences are based on two-sided tests with *p* < 0.05 taken to be statistically significant.

## 3. Results

Table 1 presents patient and caregiver characteristics, as well as their bivariate associations with changes in patient and caregiver grief (described in the next paragraph, below). A minority (42.9%) of patients were age 65 or older. Patients were majority White (61.9%), female (54.1%), and educated beyond high school (63.3%). A smaller minority (35.7%) of caregivers were 65 or older. Caregivers were also majority White (66.0%), female (67.0%), and educated beyond high school (71.4%). At baseline, 24.2%, 34.0%, and 51.0% of patients reported that they had completed a DNR order, completed an LW, or designated a healthcare proxy, respectively.

As displayed in Table 1, mean times between baseline and follow-up assessments of grief were 70.2 (*SD* = 17.0) and 73.8 (*SD* = 16.9) days, respectively, for patients and caregivers. Changes in patient grief were not found to be significantly different from zero (*n* = 98, mean = −0.03, *SD* = 0.75; *t*(97)=−0.36, *p* = 0.720), indicating that on average patient grief did not change over time. Changes in caregiver grief were significantly less than zero (*n* = 98, mean = −0.32, *SD* = 0.84; *t*(97)= −3.82, *p* < 0.001), indicating that on average caregiver grief declined over time. Changes in patient and caregiver grief were significantly, positively correlated (*n*= 98, *r* = 0.28, *p* = 0.006) with each other. Increases in patient grief were significantly associated with caregivers residing with patients (*n* = 97, *r* = 0.23, *p* = 0.022) and patient LW completion (*n* = 97, *r* = 0.23, *p* = 0.021). Decreases in caregiver grief were significantly associated with patient DNR order completion (*n* = 95, *r* = −0.22, *p* = 0.031).

Table 2 presents results from a multiple linear regression model for temporal change in patient grief (*n* = 96) that included patient insurance status, whether the caregiver resides with the patient, patient completion of an LW, and change in caregiver grief as independent variables. Having a caregiver who resides with the patient (β = 0.45, SE = 0.16; *t*(91) = 2.86, *p* = 0.005), patient completion of an LW (β = 0.48, SE = 0.14; *t*(91) = 3.29, *p* = 0.001), and change in caregiver grief (β = 0.28, SE = 0.08; *t*(91) = 3.40, *p* = 0.001) were each independently and significantly predictive of temporal increases in patient grief.

Table 3 presents results from a multiple linear regression model for temporal change in caregiver grief (*n* = 95) that included patient insurance status, patient completion of a DNR order, and change in patient grief as independent variables. Patient completion of a DNR order was significantly predictive of temporal decreases in caregiver grief (β = −0.47, SE = 0.19; *t*(91) = 2.43, *p* = 0.017). Change in patient grief was significantly, positively associated with temporal change in caregiver grief (β = 0.31, SE = 0.11; *t*(91) = 2.78, *p* = 0.007).

## 4. Discussion

The impending death of a family member with advanced cancer evokes grief in patients and family caregivers [8], yet little is known about how grief changes over time for patients and caregivers, the relationship between their grieving, and the salient predictors of grief in patients and caregivers. Using data from our longitudinal, prospective cohort study of advanced cancer patients and their family caregivers, we found that grief in patients remained relatively stable over months approaching the patient’s death, while caregivers’ grief was shown to decrease over time. In contrast to Kubler-Ross, who studied the dying process in terminally ill patients ranging from hours to several months before death [24], we find that patients’ grief did not resolve for the patient over time; however, consistent with Kubler-Ross, caregivers’ reduction in grief over time points to the possibility of grief resolution in caregivers.

In our examination of independent predictors of changes in patient and caregiver grief over time, there was evidence suggestive of synchronicity between patient and caregiver dyads’ grief, such that increases in caregiver grief was significantly, positively related to increases in patient grief, and vice versa. These results are consistent with other work demonstrating the interdependency of mental health in patient-caregiver dyads [27,28]. Further, we also found that patients who live in the same household as their caregiver, which may serve as a proxy indicator for relational closeness, experience increased grief over time. Cohabitation of the patient-caregiver dyad may elicit concerns in the patient about being a burden on the caregiver. Additionally, it may also signal attachment and dependency on the caregiver, and cognitive acceptance of the illness may heighten awareness of additional losses, including altered relationship dynamics and the need to reorganize bonds, to the forefront [14]. This is consistent with Bowlby’s assertion that grief emerges as a reaction to the disruption of strong relational bonds [13,21,22,27,39].

Our last aim examined the role of ACP on grief changes in patients and caregivers. We found that advance directive completion was, indeed, associated with reduced grief in caregivers and increased grief in patients over time. Interestingly, we did not identify any significant relationships between sociodemographic characteristics and grief, while engagement in ACP did seem to influence the grieving process in both patients and caregivers. Our results suggest that not only does grief of the patient not decline as death draws near, but that conscious preparation of advance directives such as completion of LWs actually may precipitate more grief for patients rather than resolve it. These findings suggest that patients and their family members are experiencing ACP in different ways. The identified increase in grief among patients with an LW may reflect the patient’s acknowledgment that death is near, precipitating what Freud referred to as “grief work” [40] in which patients confront their future mortality in conjunction with past and present losses [41].

Advance directive completion may help facilitate terminal illness acknowledgment, making patients more cognizant of planning and coming to terms with their own death. In fact, cognitive acceptance has been associated with DNR order completion and plays a fundamental role in end-of-life decision-making and care [41]. Although cognitive acceptance represents a patient’s acknowledgment of their terminal illness, it is reasonable to hypothesize that this recognition may lead to distress among patients confronting a myriad of physical, psychological, and social losses. Cognitive and emotional acceptance, although related, represent distinct phenomena [26] and do not appear to be achieved in tandem. Instead, grief symptoms may manifest as a byproduct of cognitive acceptance, reflecting an emotional inability to accept the loss. This is consistent with work indicating that cancer patients who cognitively accepted their impending death had more difficulty with emotional acceptance [26]. Stated another way, patients who confront the fact that they are dying might be more able to grieve their impending death.

In caregivers, we showed that changes in grief declined over time, a finding more consistent with the stages of grief progression (i.e., acceptance) initially proposed by Kubler-Ross [24]. Grief in advance of a loved one’s death has been identified as a risk factor for poor death preparedness and post-loss bereavement adjustment [14,42], with roughly 16–23% of caregivers reporting poor death preparedness [9]. Adequate death preparedness encompasses emotional (feeling at peace with the impending death), pragmatic (post-loss arrangements), and informational (medical treatment at end-of-life) domain preparation [43]. In line with prior work indicating that caregivers of patients who engage in advance care planning had higher death preparedness [9], we found that DNR order completion was associated with reductions in caregiver grief over time. These findings suggest that engagement in ACP may be somewhat therapeutic for caregivers, increasing mental preparation for what lies ahead, which may in turn foster better post-loss bereavement outcomes. Bereavement studies have proposed a theory of relief/stress reduction for caregivers, in which the reduced responsibility and reliance on caregivers following death may improve mental health [44]. However, engagement in ACP may also provide a similar sense of relief, allowing caregivers to recognize and emotionally prepare for the impending loss while reducing the decisional burden placed on caregivers on behalf of the patient at the end-of-life, all of which may promote more positive bereavement adjustment.

Taken as a whole, the identified differences in grief trajectories based on the completion of LWs in patients versus DNR orders in caregivers point to the possibility that there may be significant variation in how various forms of ACP are differentially perceived and interpreted by patients and caregivers. For patients, completion of an LW requires substantial consideration and decision-making surrounding various options for life support and end-of-life measures (e.g., artificial administration of food and water) [45] and requires the signature of the patient as well as two witnesses for the document to be considered legally binding. Thus, contemplating and ruminating about one’s end-of-life wishes and desires, and determining what constitutes quality of life in regard to their health and comfort at the end-of-life, may prompt the beginning of the grieving process in patients. Caregivers’ reduced grief as a result of DNR order completion, alternatively, may enhance the mental and emotional preparation of the patients’ impending death.

Overall, the results of this study highlight the need for effective interventions that address and facilitate the grieving process among patients and family members to foster better functioning, support, and ease patient suffering at the end of life [12], as well as to protect against poor bereavement adjustment. The American Society of Clinical Oncology [46] urges oncologists to initiate individualized ACP conversations throughout the disease course, while grief is rarely assessed by formal care providers prior to and after death. Using a family-centered care approach, patients and their caregivers should be screened for grief and psychological distress throughout the disease course, which can be accomplished by oncology and palliative care clinicians attending to the patient. Additionally, there is a need for greater recognition of and attention to the role of family and friends, as well as spiritual and religious beliefs and communities, in supporting grieving patients and their family members. The identification of grief and distress in caregivers and patients increased awareness of the complex, individualized trajectories of grief, and increased understanding of the role of ACP on grief could facilitate the provision of immediate support, which can prevent further adverse outcomes and foster well-being before, during, and following the loss of a loved one.

## 5. Strengths and Limitations

This study captured the temporal changes in grief among patients and caregivers near the end of life. It also identified the influence of advance care directives on grief in advanced cancer patients and their family caregivers. However, the study’s findings should be considered in the context of its limitations. The sample size, although sufficient to identify statistically significant changes in grief, was small, and future research should seek to replicate these findings in larger samples with more than a single repeated assessment to capture more granular changes in grief resolution over time. There is also a need to include a more comprehensive assessment of grief than that available for use in this report. We intentionally limited the number of items to assess grief for the dying patients and their family caregivers, but future research should use a more comprehensive assessment to confirm the relationships observed in this report. Particular attention to how grief trajectories may vary based on racial/ethnic diversity as well as different forms of advance directive completion should be considered in future research. Another limitation of note is the correlational nature of the study; thus, future work should focus on examining potential third variables, such as declines in health status and patient-caregiver dyad relationship quality, that may influence completion of advance care directives and changes in grief. Further, our sample consisted of caregivers and patients with cancer; future research should examine changes in grief in patient-caregiver dyads of other terminal illnesses and other modes of death. Lastly, there is a need to examine how grief relates to openness to engage in end-of-life discussions apart from completion of DNR orders or living wills and how advance care planning, including end-of-life discussions, may affect patient and family caregiver grief [47].

## 6. Conclusions

This prospective, longitudinal study examined changes in the grief experience for both advanced cancer patients and their family caregivers and found that patients’ grief remained stable, while caregivers’ grief declined over time. When examining independent predictors of grief, we found that patients who cohabitate with their caregivers show increases in grief over time. Additionally, our findings also provide support for the notion of “synchronous grieving” in patients and caregivers, such that changes in grief in one member of the dyad, or “care pair,” is associated with changes in grief in the other member and vice versa. Finally, when investigating the role of ACP on the grieving process, we found that LW completion was associated with increases in patient grief, while grief decreased in caregivers of patients who completed a DNR order, suggesting that different ACP approaches may prompt the beginning of grief work in patients while promoting the resolution of grief in caregivers.

## Figures and Tables

**Table 1 cancers-13-01977-t001:** Patient and caregiver characteristics, patient advance directives, and their associations with temporal changes in patient and caregiver grief (*n* = 98).

Patient and Caregiver Charteristics	Total Sample	No. Participants in Group	Percentage of participants in Each Group	Correlation with Change in Patient Grief		Correlation with Change in Caregiver Grief	
Patient characteristic	*N*	*n*	%	*r*	*p*	*r*	*p*
Age, 65 years or older	98	42	42.9%	−0.01	0.959	−0.05	0.655
Sex, female	98	53	54.1%	−0.01	0.952	0.07	0.516
Race, Caucasian	97	60	61.9%	0.02	0.817	−0.11	0.275
Ethnicity, Latino	98	28	28.6%	−0.04	0.714	0.05	0.643
Household income, less than 66k USD	86	49	57.0%	0.02	0.889	−0.03	0.755
Education, beyond high school	98	62	63.3%	0.11	0.263	0.07	0.514
Marital status, married	97	71	73.2%	0.06	0.565	0.09	0.383
Insurance status, insured	98	91	92.9%	−0.19	0.065	−0.19	0.066
Caregiver characteristic	*N*	*n*	%	*r*	*p*	*r*	*p*
Age, 65 years or older	98	35	35.7%	0.09	0.358	−0.02	0.811
Sex, female	97	65	67.0%	−0.16	0.119	0.00	0.966
Race, Caucasian	97	64	66.0%	0.04	0.733	−0.14	0.187
Ethnicity, Latino	98	27	27.6%	0.00	0.984	0.01	0.907
Household income, less than 66k USD	89	41	46.1%	−0.03	0.777	−0.01	0.950
Education, beyond high school	98	70	71.4%	−0.04	0.672	−0.15	0.151
Marital status, married	98	76	77.6%	−0.09	0.404	0.13	0.196
Relationship to patient, spouse	98	61	62.2%	0.12	0.230	0.01	0.906
Lives with patient, yes	97	73	75.3%	0.23	0.022	−0.02	0.860
Patient advance directive	*N*	*n*	%	*r*	*p*	*r*	*p*
DNR order, yes	95	23	24.2%	0.13	0.217	−0.22	0.031
Living will, yes	97	33	34.0%	0.23	0.021	−0.14	0.178
Healthcare proxy, yes	98	50	51.0%	0.09	0.367	−0.06	0.537
Change in grief	*N*	mean	SD	*r*	*p*	*r*	*p*
Change in patient grief	98	−0.03	0.75	1.00	N/A	0.28	0.006
Change in time, days	98	70.2	17.0	0.07	0.492	0.11	0.300
Change in caregiver grief	98	−0.32	0.84	0.28	0.006	1.00	N/A
Change in time, days	98	73.8	16.9	0.10	0.320	0.12	0.239

Notes: *N* = total number participants in the analytic sample; *n* = number of participants belonging to each category. No evidence that temporal change in patient grief is different from zero {*t*(97) = −0.36, *p* = 0.720}. Temporal change in caregiver grief is significantly different from zero {*t*(97) = −3.82, *p* < 0.001}.

**Table 2 cancers-13-01977-t002:** Results for multiple linear regression model for temporal change in patient grief (*n* = 96).

Independent Variable	Model Estimates for Change in Patient Grief				
	β	SE	*t*	*df*	*p*
Intercept	−0.05	0.28	−0.17	91	0.865
Patient insured, Y/N	−0.42	0.27	−1.57	91	0.119
Caregiver lives with patient, Y/N	0.45	0.16	2.86	91	0.005
Patient living will, Y/N	0.48	0.14	3.29	91	0.001
Change in caregiver grief	0.28	0.08	3.40	91	0.001

Notes: For binary (Y/N = Yes/No; coded 1/0) independent predictor variables, b represents the difference between “yes” and no” categories in change-over-time in patient grief. For change-over-time in caregiver grief as a predictor, b represents the increase in change-over-time in patient grief associated with one unit increase in change-over-time in caregiver grief. *t* = *t*-test statistic, *df* = degrees of freedom, *p =* significance, values at *p* < 0.05 are considered significant.

**Table 3 cancers-13-01977-t003:** Results for multiple linear regression model for temporal change in caregiver grief (*n* = 95).

Independent Variable	Model Estimates for Change in Caregiver Grief				
	β	SE	*t*	*df*	*p*
Intercept	0.09	0.30	0.30	91	0.765
Patient insured, Y/N	−0.32	0.32	−0.99	91	0.326
Patient DNR order, Y/N	−0.47	0.19	−2.43	91	0.017
Change in patient grief	0.31	0.11	2.78	91	0.007

Notes: For binary (Y/N = Yes/No; coded 1/0) independent variables, b represents the difference between “yes” and “no” categories in change-over-time in caregiver grief. For change-over-time in patient grief as a predictor, b represents the increase in change-over-time in caregiver grief associated with one unit increase in change over time in patient grief. *t* = *t*-test statistic, *df* = degrees of freedom, *p =* significance, values at *p* < 0.05 are considered significant.

## Data Availability

Data are neither finalized nor currently available to the public. Data sharing is not applicable to this article.

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
