# Peer review of "The Role of Advance Care Planning in Cancer Patient and Caregiver Grief Resolution: Helpful or Harmful?"

_cancers, 2021, doi:10.3390/cancers13081977_

Round 1

Reviewer 1 Report

Thank you for the opportunity to review this manuscript.

The study design appears appropriate however, I suggest a little more clarity needed in relation to the study sample for this study reported and relationship to the larger study.  While this information is provided, it is hard for the reader to understand with absolute clarity.

I wonder if the authors find any connection to grief and actually level of engagement with ACP - and if APC was a once off conversation or a continual one if grief was impacted.

In order to make the manuscript of interest to a wider audience, it would be worth considering the above point in light of EAPC recommendations for ACP. Reference below

Judith A C Rietjens, Rebecca L Sudore, Michael Connolly, Johannes J van Delden, Margaret A Drickamer, Mirjam Droger, Agnes van der Heide, Daren K Heyland, Dirk Houttekier, Daisy J A Janssen, Luciano Orsi, Sheila Payne, Jane Seymour, Ralf J Jox, Ida J Korfage,
Definition and recommendations for advance care planning: an international consensus supported by the European Association for Palliative Care,
The Lancet Oncology,
Volume 18, Issue 9,2017, Pages e543-e551,https://doi.org/10.1016/S1470-2045(17)30582-X.

Author Response

Reviewer #1:

Comment #1: The study design appears appropriate however, I suggest a little more clarity needed in relation to the study sample for this study reported and relationship to the larger study.  While this information is provided, it is hard for the reader to understand with absolute clarity.

Response #1: We now note in the Methods section that this is a secondary analysis of a multi-site prospective longitudinal cohort study designed to examine Latinx versus non-Latinx disparities in end-stage cancer care. We had written: “The sample for the present study (N=98 patient-caregiver dyads) is a subsample of patient and caregiver participants in Coping with Cancer III (CwC III), a multi-site, prospective, longitudinal cohort study of advanced cancer patients and their caregivers funded by the National Institute of Minority Health and Health Disparities (MD007652; MPIs: Maciejewski/ Prigerson) to examine Latino/non-Latino disparities in ACP and end-of-life care.” On lines 103-105 we now have added the following statement to clarify the study sample: “Thus, this is a secondary analysis of these data for the purpose of examining how grief changes in patients and their family caregivers as the patient approaches death and the influence of advance care planning on grief resolution of both patients and their family caregivers.

Comment #2: I wonder if the authors find any connection to grief and actually level of engagement with ACP - and if APC was a once off conversation or a continual one if grief was impacted.

Response #2: We should clarify that our study examined completion of an advanced directive such as a Living Will or a Do Not Resuscitate order (i.e., discrete acts) and was not about end-of-life discussions or the process of conversing about these issues over time. We do and have had a keen interest in end-of-life discussions as an important process and now reference the EAPC recommendations for ACP citing Rietjens et al. 2017, but that is not what we were examining in this analysis which was a clear dichotomy between completing these advance directives or not at the two assessed timepoints. On lines 331-334 of the revised manuscript we now write: “Lastly, there is a need to examine how grief relates to an openness to engage in end-of-life discussions apart from completion of DNR orders or Living Wills and how advance care planning, including end-of-life discussions, may affect patient and family caregiver grief [48].” Citation #48 is the Rietjens et al. 2017 article.

Reviewer 2 Report

The manuscript is well written, the communication is excellent, the study is interesting and covers a highly needed topic. Conclusion are well elaborated. On the other hand, more literature should be presented about the advance care planning (ACP) and grief resolution among cancer patients and caregivers. Authors correctly explain the lack of literature in this field, but they should present a state of-art of this topic. 

Among the limits of their study it should be mentioned that the instrument they administered was rather limited,, though pertinent. 

It would have been interesting to know more about the support received by relatives and friends , in relation to the evolution of the grief. Also the role of relies faith may play a role in the elaboration of grief. This point might be added for further development of this study

Author Response

Reviewer #2:

Comment #1: The manuscript is well written, the communication is excellent, the study is interesting and covers a highly needed topic. Conclusion are well elaborated. On the other hand, more literature should be presented about the advance care planning (ACP) and grief resolution among cancer patients and caregivers. Authors correctly explain the lack of literature in this field, but they should present a state of-art of this topic. 

Response #1: We appreciate this comment and agree that we are challenged to identify an article on how ACP relates to patient and caregiver grief.  As noted above, we focus on DNR and Living Will completion, which are discrete events and not the process of ACP including discussions of end-of-life wishes.  Also as noted above, we discuss the need for more research on how ACP relates to grief resolution in patients and their family caregivers (e.g., see response #2 to Reviewer #1).

Comment #2: Among the limits of their study it should be mentioned that the instrument they administered was rather limited, though pertinent. 

Response #2: We now further acknowledge that a limitation of this study is that the grief assessment was a reduced set of items from our PG-12.  We did this intentionally because patients were terminally ill and wanted to reduce burden on them.  Nevertheless, we agree that a fuller assessment would provide a stronger empirical basis for the conclusions of the relationships reported in this study. On lines 322-325 we now write: “There is also a need to include a more comprehensive assessment of grief than that available for use in this report. We intentionally limited the number of items to assess grief for the dying patients and their family caregivers, but future research should use a more comprehensive assessment to confirm the relationships observed in this report.

Comment #3: It would have been interesting to know more about the support received by relatives and friends in relation to the evolution of the grief. Also the role of relies faith may play a role in the elaboration of grief. This point might be added for further development of this study

Response #3: We are delighted that Reviewer #2 raised this issue as we are developing a social theory of grief that highlights this very point – the crucial role of friends and family in bereavement adjustment and specifically grief resolution.  We also are developing healthcare chaplain interventions in outpatient oncology clinics to address the spiritual needs of advanced cancer patients and their family caregivers. On lines 309-311 we now write: “Additionally, there is a need for greater recognition of and attention to the role of family and friends, as well as spiritual and religious beliefs and communities in supporting grieving patients and their family members.